# Single cell transcriptomics reveals opioid usage evokes widespread suppression of antiviral gene program

Tanya T. Karagiannis [1,2,9], John P. Cleary Jr [2,3,9], Busra Gok [2,4], Andrew J. Henderson[5], Nicholas G. Martin [6], Masanao Yajima [7], Elliot C. Nelson [8] & Christine S. Cheng[1,2,3,4 ✉]

Chronic opioid usage not only causes addiction behavior through the central nervous system, but also modulates the peripheral immune system. However, how opioid impacts the immune system is still barely characterized systematically. In order to understand the immune modulatory effect of opioids in an unbiased way, here we perform single-cell RNA sequencing (scRNA-seq) of peripheral blood mononuclear cells from opioid-dependent individuals and controls to show that chronic opioid usage evokes widespread suppression of antiviral gene program in naive monocytes, as well as in multiple immune cell types upon stimulation with the pathogen component lipopolysaccharide. Furthermore, scRNA-seq reveals the same phenomenon after a short in vitro morphine treatment. These findings indicate that both acute and chronic opioid exposure may be harmful to our immune system by suppressing the antiviral gene program. Our results suggest that further characterization of the immune modulatory effects of opioid is critical to ensure the safety of clinical opioids.

[1] Program in Bioinformatics, Boston University, 24 Cummington Mall, Boston, MA 02215, USA. [2] Department of Biology, Boston University, 5 Cummington Mall, Boston, MA 02215, USA. [3] Program in Molecular Biology, Cell Biology and Biochemistry, Boston University, 24 Cummington Mall, Boston, MA 02215, USA. [4] Program in Cell and Molecular Biology, Boston University, 24 Cummington Mall, Boston, MA 02215, USA. [5] Department of Medicine and Microbiology, Boston University School of Medicine, 650 Albany St, Boston, MA 02215, USA. [6] QIMR Berghofer Medical Research Institute, Brisbane, QLD 4006, Australia. [7] Department of Mathematics and Statistics, Boston University, 111 Cummington Mall, Boston, MA 02215, USA. [8] Department of Psychiatry, Washington University School of Medicine, 660S. Euclid Ave, St. Louis, MO 63110, USA. [9] These authors contributed equally: Tanya T. Karagiannis, John P. Cleary Jr. ✉email: chcheng@bu.edu

The opioid epidemic is a major threat to global public health that affects millions of people and their families. Part of the problem is caused by the rapid increase in the number of opioid prescriptions written by medical practices starting from the late 1990s. From 1999 to 2017, overdoses related to prescription opioids have dramatically increased in the United States with overdose deaths found to be five times higher in 2017 compared to 1999 (ref. [1]). In addition, opioids affect not only the central nervous system (CNS) but also the peripheral immune system through the expression of a variety of opioid receptors on different immune cell types[2]. However, the effect of opioids on the peripheral immune system is complicated and involves various mechanisms. Studies have shown inconsistent results, where some suggest opioid usage is immunosuppressive while, in contrast, others suggest opioids are immunoactivating[2–4]. Most of these studies focus on a particular immune cell subpopulation and a few candidate genes. Interestingly, epidemiological studies suggest that opioid usage is associated with increased susceptibility to opportunistic infections such as tuberculosis, HIV, and pneumonia[5–7]. Given the extensive use of prescription opioids and the global opioid epidemic, it is important to understand how opioid usage modulates different cell types in the immune system.

Next-generation sequencing technologies such as RNA sequencing (RNA-seq) have become standard for querying gene expression in tissues and cells. Yet gene expression levels obtained through such ensemble-based approaches generate expression values averaged across large populations of cells, masking cellular heterogeneity. Primary cells such as peripheral blood immune cells or tissue samples from patients usually comprise heterogeneous cell populations. It is therefore highly time consuming and labor intensive to separate and study the individual cell types and generally not feasible given the limited input material from patient samples. Recent experimental advances have allowed the isolation of RNA from a single cell and the generation of cDNA libraries that can be sequenced from small amounts of RNA[8].

Here, using single-cell RNA sequencing (scRNA-seq), we are now able to determine expression profiles in single-cell resolution. ScRNA-seq is a powerful tool to identify and classify distinct cell populations, characterize rare subpopulations, and trace cells along dynamic cellular stages, such as during differentiation or disease progression[9]. The recent development of massively parallel microdroplet-based scRNA-seq approaches allows profiling of gene expression of thousands to millions of single cells from a limited quantity of sample at a reduced cost[10,11]. Here, we utilize microdroplet-based scRNA-seq to systematically characterize cell-type specific gene expression in the peripheral immune system of opioid-dependent individuals compared to non-dependent controls.

## Results

### Peripheral blood mononuclear cell scRNA-seq of opioid users shows suppression of antiviral genes.

Using microdroplet-based scRNA-seq, we profiled gene expression in 57,271 single cells from the peripheral blood mononuclear cells (PBMCs) of seven opioid-dependent individuals and seven age/sex-matched non-dependent controls (averaging 3980 single cells per individual) (Fig. 1a). To examine opioid usage-specific changes in gene expression in response to pathogenic stimuli, we treated PBMCs from three of the seven opioid-dependent individuals and three of the controls with lipopolysaccharide (LPS, a component of Gram-negative bacteria) for 3 h and profiled 22,326 single cells. We sequenced these single cells to an average depth of 21,801 reads per cell and detected on average 805 genes and 2810 transcripts per cell. To identify each of the immune cell subpopulations, we applied dimensionality reduction methods, including principal component analysis (PCA) and t-stochastic neighbor embedding (t-SNE), and unsupervised graph-based clustering[12,13]. We identified 12 immune cell types/states using expression of canonical gene markers (Fig. 1b, Supplementary Figs. 1–3). Of the naive state immune populations, we identified CD4+ T cells, naïve CD8+ T cells, memory CD8+ T cells, natural killer (NK) cells, B cells, and monocytes. Of the LPS-treated immune populations, we identified CD4+ T cells, CD8+ T cells, activated T cells, B cells, NK cells, and monocytes (Fig. 1b, Supplementary Figs. 1–3). We observed a slight shift in global gene expression states between opioid dependent and control samples in most of the naïve cell types, while we observed larger differences in gene expression states in LPS-treated cell types (Fig. 1b).

From differential gene expression analysis for each immune subpopulation, we observed a downregulation of interferon-stimulated genes and antiviral genes in opioid-dependent samples compared to control samples. The suppression of antiviral genes was observed only in monocytes in naive state and in most immune cell subpopulations under LPS treatment (Fig. 1c, Supplementary Figs. 4 and 5). CD8+ T cells under LPS treatment exhibited suppression of antiviral genes to a lesser degree (Supplementary Fig. 5). This was further confirmed with pathway enrichment analysis of the resulting differential genes. We observed higher enrichment of defense response to virus and interferon signaling pathways in monocytes in naïve state and in most of the immune cell subpopulations in LPS-treated state in control samples (Fig. 1d). LPS activates several innate immune response transcriptional modules: core antiviral response, peaked and sustained inflammatory genes that were previously characterized in mouse bone marrow-derived dendritic cell cultures[14] (Fig. 2a, Supplementary Table 1). We found widespread suppression of antiviral genes in opioid-dependent cells across LPS-treated immune subpopulations while peaked and sustained inflammatory genes were modestly affected by opioid usage (Fig. 2b, Supplementary Figs. 8–12). Taken together, our data suggest that chronic opioid usage results in widespread suppression of antiviral genes affecting all immune subpopulations including both innate and adaptive immune cell types.

To examine if the observed suppression of the antiviral gene program upon LPS treatment in opioid-dependent individuals is affected specifically through the *TLR4* receptor pathway by inactivation of the *TRIF* signaling cascade, we seek an alternative way to activate type I interferon pathway directly. We activated the antiviral gene program with interferon beta (*IFNβ*), which directly activates type I interferon response and antiviral gene program through the interferon alpha and beta receptors. Given that interferon alpha and beta receptor subunits 1 and 2 (*IFNAR1* and *IFNAR2*) are expressed in all immune cell populations (Supplementary Fig. 13), we expect the activation of the antiviral gene program and interferon response genes in each of these immune subpopulations upon *IFNβ* treatment. In order to perform scRNA-seq in a cost-effective way and also to reduce technology driven batch effects, we performed scRNA-seq with an antibody-based cell-hashing technique to multiplex samples in droplet-based scRNA-seq[15] (Supplementary Fig. 14; see Methods). We profiled 9278 single PBMCs treated with *IFNβ* for 3 h from three opioid-dependent individuals and three age/sex-matched non-dependent controls (averaging 1547 single cells per individual) (Supplementary Fig. 14). We observed that activation of the antiviral gene program is at the same level between opioid-dependent individuals and non-dependent controls in each of the cell types (Supplementary Fig. 15). Our results suggest that the suppression of the antiviral gene program in opioid-dependent cells is a stimulus-specific phenotype that is most likely affected through the *TLR4* pathway.

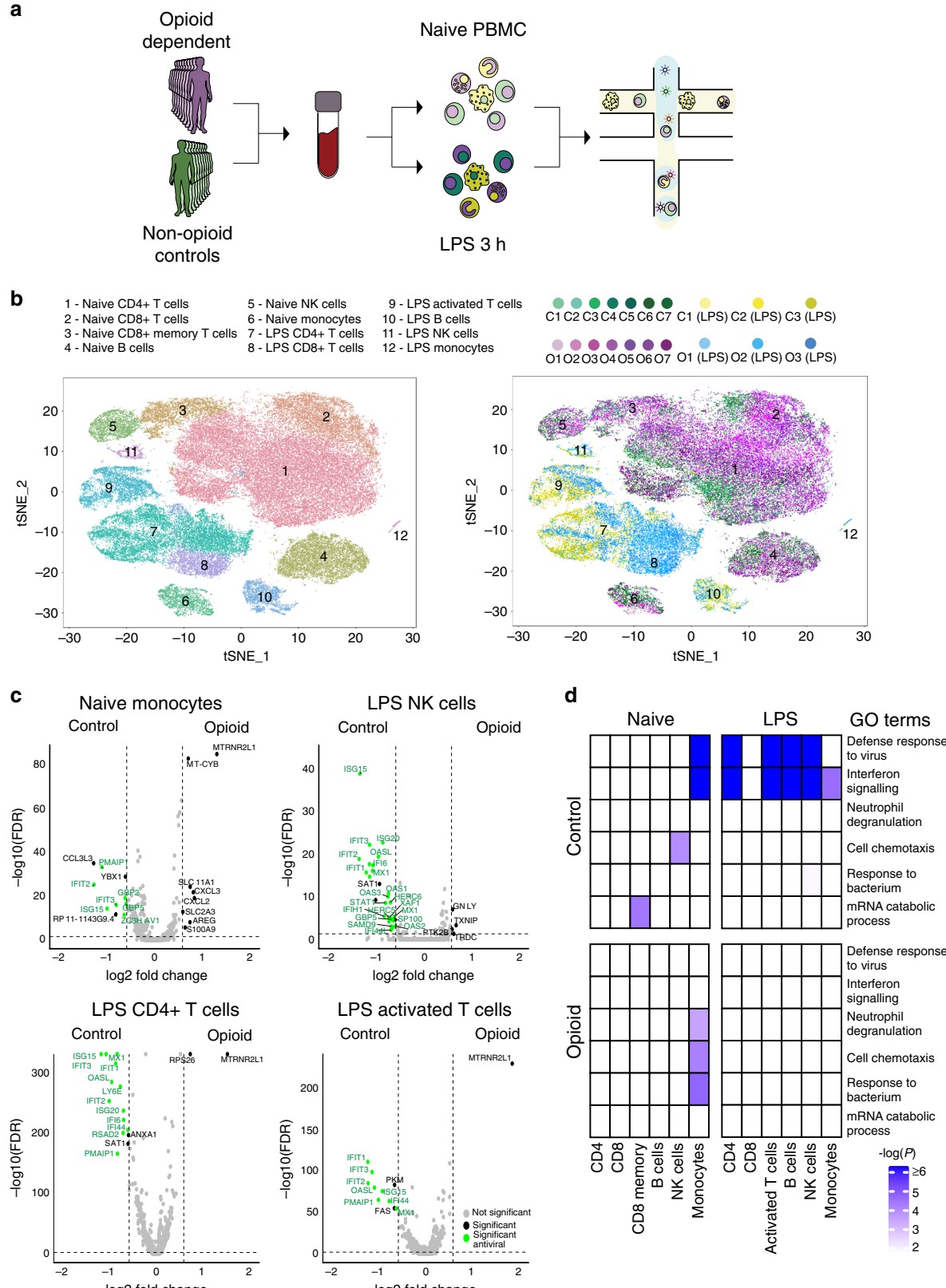

**Morphine reduces antiviral genes in LPS-treated PBMC**. To examine the in vitro effect of opioids, we first treated primary human PBMCs from healthy individuals with a titration of morphine for 24 h before stimulating with either a mock treatment (Untreated) or 100 ng/mL LPS for 3 h. We then performed quantitative reverse transcription PCR (RT-qPCR) using primers against the major antiviral gene, *ISG15*, which was the most prominent antiviral gene downregulated in opioid-dependent cells across cell types. We found that PBMCs pretreated with morphine for 24 h exhibited a dose-dependent inhibition to the induction of *ISG15* after LPS treatment (Fig. 3a). Furthermore, this inhibition was also detectable after only 3 h of morphine

**Fig. 1 scRNA-seq revealed a widespread suppression of antiviral genes in opioid-dependent individuals. a** Experimental workflow schematic. Peripheral blood from opioid-dependent individuals and control individuals were collected, PBMCs were isolated, and microdroplet-based scRNA-seq was performed using Chromium Controller (10X Genomics). **b** t-SNE plot of naive (51,041) and LPS (100 ng/mL)-treated (21,873) PBMCs were clustered (cells were filtered based on >300 and <2000 genes per cell, <10,000 UMIs per cell; see Methods) and identified into immune populations (top) and visualized by control individuals and opioid-dependent individuals in each state (bottom): naive state control samples 1–7 (C1–C7), naive state opioid-dependent samples 1–7 (O1–O7), LPS-treated control samples 1–3 (C1–C3 (LPS)), LPS-treated opioid-dependent samples 1–3 (O1–O3 (LPS)). **c** Volcano plot showing fold change of gene expression (log2 scale) for downregulated (Control) and upregulated (Opioid) genes for opioid-dependent cells compared to non-dependent controls for naive state populations: monocytes and LPS-treated populations: NK cells, CD4+ T cells, and activated T cells (x-axis) with a significance of 0.05 (y-axis, −log10 scale). Significant genes shown with black dots, significant antiviral genes shown with green dots, and insignificant genes shown with gray dots. **d** Pathway enrichment analysis of significant differential genes across all naive and LPS-treated cell types evaluated by −log10(p value) as indicated by blue-purple scale (x-axis: cell type/state, y-axis: pathways). White represents an analysis which did not provide enrichment results for the specific pathway. Source data listing genes and expression values for **c** and **d** are provided in Source Data file. Similar findings were observed in repeat experiments using different patient samples.

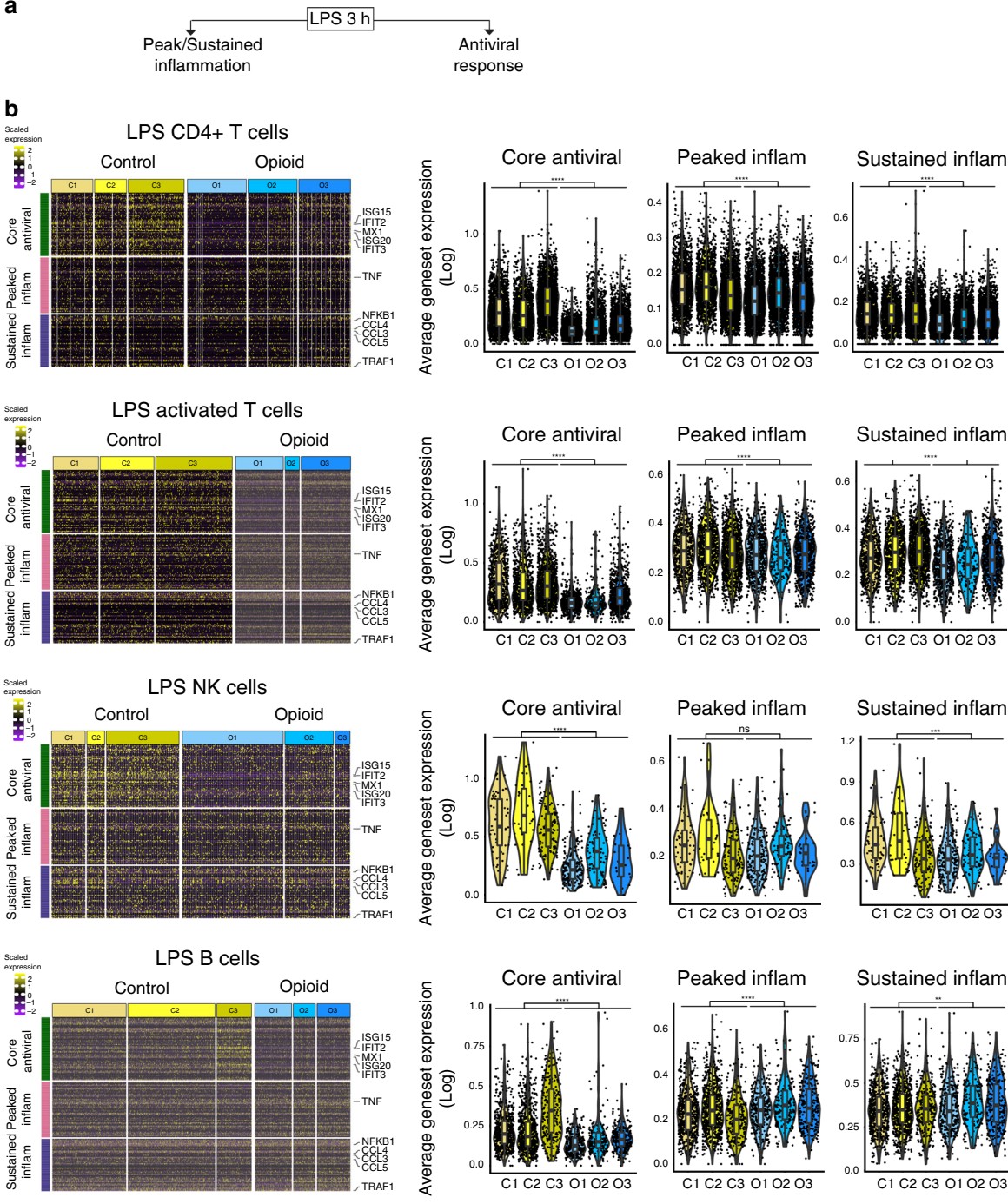

**Fig. 2 LPS-stimulated antiviral gene program were consistently suppressed in opioid-dependent individuals. a** Evaluation of the three innate immune response gene programs stimulated by LPS: antiviral, peaked inflammatory, and sustained inflammatory. **b** Left: Heatmap of scaled expression of core antiviral and inflammatory response genes observed in control sample cells (C1–C3) and opioid-dependent sample cells (O1–O3) in LPS-treated populations: CD4+ T cells, activated T cells, NK cells, and B cells. Color scale for heatmap indicates scaled gene expression. Yellow indicates positive scaled gene expression, purple indicates negative scaled gene expression, and while black represents zero scaled gene expression. Labeled key antiviral and inflammatory genes expression across LPS sample cells in the LPS-treated populations (Supplementary Figs. S6 and S7). Right: Average expression of all genes in a geneset (log expression) for each cell, grouped by control samples (C1–C3) and opioid-dependent samples (O1–O3) for LPS-treated populations: CD4+ T cells (C1–C3: 5211 cells and O1–O3: 6378 cells), activated T cells (C1–C3: 2456 cells and O1–O3: 1578 cells), NK cells (C1–C3: 268 cells and O1–O3: 351 cells), and B cells (C1–C3: 1527 cells and O1–O3: 747 cells). Inset box plots show the median, lower and upper hinges that correspond to the first quartile (25th percentile) and third quartile (75th percentile), and the upper and lower whiskers extend from the smallest and largest hinges at most 1.5 times the interquartile range. For CD4+ T cells, two-tailed $T$-test with comparison tests between control and opioid-dependent groups for each geneset: core antiviral ($p < 2.22e{-}16$), peaked inflammation ($p < 2.22e{-}16$), sustained inflammation ($p < 2.22e{-}16$). For activated T cells, two-tailed $T$-test with comparison tests between control and opioid-dependent groups for each geneset: core antiviral ($p < 2.22e{-}16$), peaked inflammation ($p = 2.7e{-}08$), sustained inflammation ($p < 2.22e{-}16$). For NK cells, two-tailed $T$-test with comparison tests between control and opioid-dependent groups for each geneset: core antiviral ($p < 2.22e{-}16$), peaked inflammation ($p = 0.44$), sustained inflammation ($p = 0.00054$). For B cells, two-tailed $T$-test with comparison tests between control and opioid-dependent groups for each geneset: core antiviral ($p < 2.2e{-}16$), peaked inflammation ($p = 3.7e{-}6$), sustained inflammation ($p = 0.0033$). $^{ns}p > 0.05$, $*p < 0.05$, $**p < 0.01$, $***p < 0.001$, $****p < 0.0001$. Similar findings were observed in repeat experiments using different patient samples.

pretreatment followed by 3 h of LPS treatment (Fig. 3b). In order to characterize this phenomenon at a genome-wide scale, we performed scRNA-seq with the cell-hashing technique and profiled 2946 single PBMCs treated with morphine alone and then treated with LPS for 3 h (averaging 740 single cells per sample) (Supplementary Fig. 16). We found a modest but consistent suppression of core antiviral genes in response to morphine exposure. This phenotype was most pronounced in CD4+ T cells, CD8+ T cells, and NK cells (Fig. 3c, Supplementary Figs. 17–21).

## Discussion

Our results show that there is widespread suppression of interferon-stimulated genes and antiviral genes in multiple innate and adaptive peripheral immune subpopulations both ex vivo and in vitro upon LPS treatment. Our findings suggest a potential adverse effect from opioid usage on the defense response towards viral infection in the immune system. This may explain in part the higher susceptibility to viral infection in opioid users observed in epidemiological studies[5–7,16]. In addition, our in vitro findings also demonstrate that the observed suppression of antiviral pathway from our ex vivo experiments does not arise from needle sharing or presence of hepatitis C virus infection in these injection opioid users. Given that most recreational opioid users inject drugs and many prescription opioid users are post-surgery or cancer patients under chemotherapy treatments, they are already more prone to infection; therefore, suppression of antiviral genes with opioid usage brings clinical relevance and demonstrates the importance of carefully examining each individual case to avoid any possibility of comorbidity.

Acute and chronic opioid use has been shown to modulate the immune system and increase risk of opportunistic infections. This is supported by both epidemiological studies and animal studies where the variable from sharing of contaminated needles can be eliminated[17]. Previous reports using in vitro cell culture model and in vivo opioid treatment rodent models have shown that opioids affect both the innate and adaptive immune function[2–4,17]. For example, morphine has been shown to reduce *IL6* and *TNFα* expression in macrophages and reduce *IL8* expression in neutrophils while demonstrating increases in Th1 cell death and Tbet activity in T cells[17]. In addition to the peripheral immune system, studies have suggested that opioids create a neuroinflammatory response in the CNS through MOR-independent pathways. Reports from Hutchinson et al.[18] and Wang et al.[19] have shown that morphine activates the TLR4 receptor through a MOR-independent pathway in glial populations in the CNS and contributes to drug reinforcement. Opioid antagonist,

β-funaltrexamine also has been shown to inhibit *NFkB* signaling and chemokine expression in human astrocytes and inhibit LPS-induced neuroinflammation in mice[20,21]. However, these studies usually focus on a few genes and in a particular cell type. Furthermore, very few of these immune function characterization studies were performed using primary immune cells directly from opioid-dependent patients. There is currently no study that provides a systems level and genome-wide view of the immune system from chronic opioid usage. Our study represents the first genome-wide and single-cell level transcriptomics study to characterize peripheral immune cell populations directly from chronic opioid users. Furthermore, we have identified stimulus-specific and cell-type-specific dysregulation of the immune response gene regulatory circuitry upon chronic opioid usage and acute opioid treatment. Our results provide potential systems level molecular explanation to the widely observed higher susceptibility to opportunistic infection in opioid using individuals from epidemiological studies.

Opioid-induced immune modulation is mainly thought to occur through opioid receptors present on peripheral immune cell types[2,22–24]. However, the presence of opioid receptors in peripheral immune cells is controversial. While several studies have shown that classical opioid receptors such as MOR are expressed on various peripheral blood immune cell types[25–30], other studies evaluated the presence of opioid receptors in PBMCs in which they failed to detect mRNA transcripts for all opioid receptors except for the nonclassical receptor NOR[4,31]. Due to the nature of single-cell assays such as scRNA-seq, the expression level of opioid receptors at a single-cell level was very low or not detectable. Therefore to clarify whether opioid receptors are present in peripheral immune cells, we looked at the expression of opioid receptors in population level RNA-seq data from PBMCs of healthy individuals from a previous study[32]. We found that the classical opioid receptor MOR is expressed in CD4+ T cells, CD8+ T cells, monocytes, and NK cells, but not in B cells, while the other two classical opioid receptors DOR and KOR are very low in expression or undetectable (Supplementary Fig. 22). The nonclassical receptor NOR is expressed in all immune cell types and is higher expressed in monocytes (Supplementary Fig. 22). We anticipate the immune modulatory effect we observed from in vivo opioid usage and in vitro opioid treatment potentially occurs through both MOR and NOR receptors.

The stimulation of toll-like receptor 4 *TLR4* by LPS induces expression of innate immune response genes previously categorized into three gene modules: antiviral, peaked inflammatory,

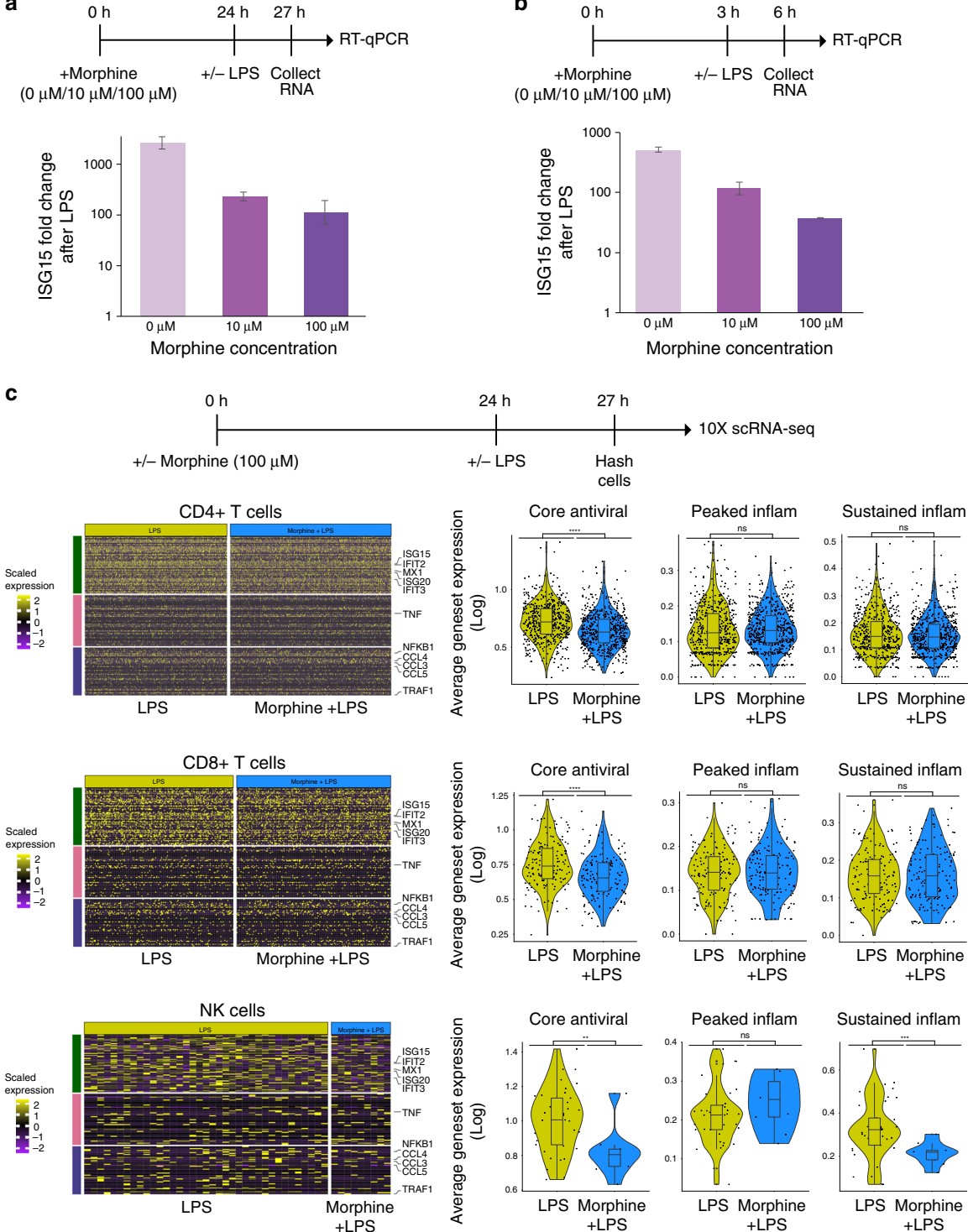

and sustained inflammatory genes[14]. Type I interferons function as autocrine and paracrine factors to induce antiviral gene activation in response to LPS[33,34]. We have observed strong suppression of the antiviral gene program in response to LPS in PBMCs of opioid-dependent individuals (Fig. 2). Although there is some evidence of *TLR4* expression in other immune cell types such as CD4+ T cells[35] and NK cells[36] in naive PBMCs, monocytes are the major cell type that express high levels of *TLR4* while other immune cell types demonstrate low expression levels as shown from reanalysis of previously published population level RNA-seq data from PBMCs of healthy individuals[32]

(Supplementary Fig. 13). We anticipate that LPS induction of the three innate immune response gene pathways by *TLR4* activation occurs mainly in monocytes; this leads to the expression of autocrine and paracrine factors such as *TNFα* and *IFNα/IFNβ* which then induce expression of the innate immune response gene modules in other immune cell types through the activation of *TNF* receptors and *IFNα/β* receptors as shown by the expression of these receptors in CD4+ T cells, CD8+ T cells, B cells, and NK cells. In addition, *IFNβ* activates the antiviral gene program directly through the *IFNAR* receptors in each of the immune cell types (Supplementary Fig. 15) given its ubiquitous

**Fig. 3 Short exposure to morphine resulted in suppression of antiviral genes upon LPS treatment. a, b** Evaluation of ISG15 mRNA expression after morphine treatment. PBMCs from a healthy, non-opioid-exposed individual were pretreated with morphine (0, 10, 100 μM) for 24 h (**a**) or 3 h (**b**) followed by LPS (100 ng/mL) stimulation for 3 h. Interferon pathway gene *ISG15* expression was evaluated by RT-qPCR. Values displayed as fold increase (log10) to gene expression in LPS-treated cells over unstimulated cells, plus or minus one standard deviation. Error bars here represent technical variability; experiments were repeated at least three times with similar results. **c** Cell hashing scRNA-seq of healthy PBMCs pretreated with morphine for 24 h followed by LPS (100 ng/mL) treatment for 3 h. Left: Heatmaps of scaled expression of core antiviral response genes observed in LPS-treated populations: CD4+ T cells, CD8+ T cells, and NK cells. Color scale for heatmap indicates scaled gene expression. Yellow indicates positive scaled gene expression, purple indicates negative scaled gene expression, and while black represents zero scaled gene expression Right: Average expression of all genes in a geneset (log expression) for each cell, grouped by mock-treated and morphine-treated cells of LPS-treated populations: CD4+ T cells (LPS (534 cells), Morphine+LPS (605 cells)), CD8+ T cells (LPS (152 cells), Morphine+LPS (158 cells)), and NK cells (LPS (37 cells), Morphine+LPS (9 cells)). Inset box plots show the median, lower and upper hinges that correspond to the first quartile (25th percentile) and third quartile (75th percentile), and the upper and lower whiskers extend at most 1.5 times the interquartile range. All comparisons use two-tailed *T*-tests. For CD4+ T cells: comparison between control and opioid-dependent groups for each geneset: core antiviral ($p < 2.22e-16$), peaked inflammation ($p = 0.91$), sustained inflammation ($p = 0.16$). For CD8+ T cells: comparison between control and opioid-dependent groups for each geneset: core antiviral ($p = 6.3e-07$), peaked inflammation ($p = 0.91$), sustained inflammation ($p = 0.85$). For NK cells: comparison between control and opioid-dependent groups for each geneset: core antiviral ($p = 0.0053$), peaked inflammation ($p = 0.23$), sustained inflammation ($p = 0.00039$). $^{ns}p > 0.05$, $^*p < 0.05$, $^{**}p < 0.01$, $^{***}p < 0.001$, $^{****}p < 0.0001$. Source data for **a** and **b** detailing expression values are provided in Source Data file.

expression (Supplementary Fig. 13 for *IFNAR1* and *IFNAR2* expression). Since we did not observe suppression of the antiviral gene program in response to direct *IFNβ* treatment (Supplementary Fig. 15), we speculate the modulatory effect of opioids are affecting a component in the the *TLR4/TRIF* signaling cascade in both the naive and LPS-treated conditions primarily in monocytes, which could explain the observed suppression of the antiviral gene program in response to LPS (Figs. 1 and 2).

Furthermore, our study demonstrates the utility of scRNA-seq as an unbiased tool to assess cell-type-specific and stimulus-specific genome-wide transcriptomic phenotype from limited quantity of patient samples. Upon a stimulation condition, such as LPS treatment, the opioid-induced phenotype is much more pronounced and resembles the signal amplifying effect from electronic amplifiers. We anticipate this type of signal amplification method coupled with single-cell transcriptomics will be of broad interest and can be applied to many other disease models where disease relevant stimuli can be used to activate naïve PBMCs isolated from patients that are otherwise quiescent to amplify signal over noisy background and thus reveal the phenotype of the disease.

Our finding of opioid-induced widespread suppression of antiviral gene program upon LPS treatment may suggest that in addition to the adverse effects of addiction behavior, opioid usage might increase susceptibility to opportunistic viral infection. Chronic prescription of opioid use is common in cancer patients, many of which are also going through chemotherapy that modulate and weaken the immune system. Our finding suggests that deeper understanding of the immune modulatory effect of opioid in the context of these clinical conditions is needed and that precaution is needed by clinicians when prescribing opioids to patient groups that are already more susceptible to opportunistic infections.

## Methods

**PBMCs from opioid-dependent individuals**. Frozen vials of PBMCs prepared from the fresh blood of opioid-dependent (mostly heroin dependent), and non-dependent neighborhood control individuals were collected in the Comorbidity and Trauma Study (CATS)[37,38] and subsequently obtained from the biorepository of National Institute on Drug Abuse (NIDA, Rockville MD). We evaluated 14 age-matched subjects ranging from 24 to 45 years in age, with an equal number of male and female subjects (Supplementary Table 6). Cases were recruited from opioid replacement therapy (ORT) clinics in the greater Sydney, Australia region, while controls were recruited from areas in close proximity to ORT clinics (neighborhood controls). Cases and controls were required to be English speakers 18 years of age or older. Cases were participants in ORT for opioid dependence while controls were excluded for recreational opioid use 11 or more times lifetime. All subjects provided written informed consent[37,38]. All samples were stripped of personally

identifying information and assigned sample ID numbers prior to receipt. No further ethical oversight was required from the Boston University IRB following de-identification of the procured samples.

**scRNA-seq of LPS-treated patient PBMCs**. Frozen PBMCs isolated from the blood of opioid-dependent and non-dependent neighborhood individuals were revived, and live cells were isolated via fluorescently activated cell sorting (FACS) using a Sony SH800 cell sorter and a live/dead cell stain (LIVE/DEAD Fixable Green Cell Stain Kit, for 488 nm Excitation, Thermo Fisher, L34969). The FACS gating strategy used for the isolation of live PBMC is illustrated in Supplementary Fig. 23. Dilutions were prepared from all 14 samples at a concentration of 1000 cells/μL as outlined in the Chromium Single Cell 3′ Reagent Kit v2 User Guide (10X Genomics, CG00052 Rev.B), and 7000 cells per sample were used to perform the droplet-based Chromium Single Cell 3′ scRNA-seq method (10X Genomics, Chromium Single Cell 3′ Library and Gel Bead Kit, Cat# PN-120237). In total, 200,000 cells from 6 of the 14 samples (three dependent and three non-dependent) were plated into a non-tissue culture-treated 96-well plate in a leukocyte-supporting complete RPMI medium (10% HI-FBS, 1% L-glutamate, 1% NEAA, 1% HEPES, 1% sodium pyruvate, 0.1% B-mercaptoethanol). Lipopolysaccharide (LPS) (Invivogen, LPS-EK Ultrapure, Cat# tlrl-pekpls) was then added to a final concentration of 100 ng/mL and the cells were incubated at 37 °C for 3 h. Cells were then collected, washed, and diluted to 1000 cells/μL before being used to perform the 10X Genomics Chromium Single Cell 3′ method as outlined in the Single Cell 3′ Reagent Kit v2 User Guide. Briefly, 20 μL of 1000 cells/μL PBMC suspension from each subject/condition were combined, and 33.8 μL of cell suspension (total cell number = 33,800) was mixed with 66.2 μL of RT reaction mix before being added to a chromium microfluidics chip already loaded with 40 μL of barcoded beads and 270 μL of partitioning oil. The chip was then placed within the chromium controller where single cells and barcoded beads were encapsulated together within oil droplets. Reverse transcription was then performed within the oil droplets to produce barcoded cDNA. Barcoded cDNA was isolated from the partitioning oil using Silane DynaBeads (Thermo Fisher Scientific, Dynabeads MyONE Silane, Cat# 37002D) before amplification by PCR. Cleanup/size selection was performed on amplified cDNA using SPRIselect beads (Beckman-Coulter, SPRIselect, Cat# B23317) and cDNA quality was assessed using an Agilent 2100 BioAnalyzer and the high-sensitivity DNA assay (Agilent, High-Sensitivity DNA Kit, Cat# 5067-4626). Sequencing libraries were generated from cleaned, amplified cDNA using the 10X Chromium Kit's including reagents for fragmentation, sequencing adaptor ligation, and sample index PCR. Between each of these steps, libraries were cleaned and size selected using SPRIselect beads. Final quality of cDNA libraries was once again assessed using the Agilent BioAnalyzer High-Sensitivity DNA assay, and quality-confirmed libraries were sequenced using Illumina's NextSeq platform. All reagents are listed in Supplementary Table 2.

**scRNA-seq of IFNβ-treated patient PBMCs**. Frozen PBMCs isolated from the blood of three opioid-dependent and three non-dependent neighborhood individuals were revived, and live cells were isolated via FACS using a Sony SH800 cell sorter and a live/dead cell stain (LIVE/DEAD Fixable Green Cell Stain Kit, for 488 nm Excitation, Thermo Fisher—L34969). The FACS gating strategy used for the isolation of live PBMC is illustrated in Supplementary Fig. 23. Live cells were plated into a non-tissue culture-treated 96-well plate in a leukocyte-supporting complete RPMI medium (10% HI-FBS, 1% L-glutamate, 1% NEAA, 1% HEPES, 1% sodium pyruvate, 0.1% B-mercaptoethanol) at a density of 200,000 cells per well. Twenty-two microliters of a 100 U/mL solution of IFNβ (Recombinant Human IFN-beta Protein, R&D Systems, Cat# 8499-IF-010) was then added to each well

(for a final IFNβ concentration of 10 U/mL) and cells were incubated for 3 h at 37 °C. After treatment, all cells were collected, and an equal number of cells per patient sample were collected, washed, and each sample was "hashed" using unique oligonucleotide-barcoded antibodies[15] (Supplementary Table 4) to track the cells' well/condition of origin. Briefly, cells were suspended in Cell Staining Buffer (BioLegend, Cat# 420201) and blocked using Human TruStain FcX reagent (Bio-Legend, Cat# 422301). Cells were then incubated with 1 μg of TotalSeq antibodies (BioLegend, Cat# 3964601, 394603, 394605, 394607, 394609, 394611, 394613, 394615, 394617, 394619, 394623, 394625), washed with PBS, and filtered through 40 μM cell strainers (Bel-Art, Flowmi Cell Strainer, Cat# H13680-0040). Samples were then normalized to 1000 cells/μL, mixed in equal measure (20 μL each), and used to perform the Chromium Single Cell 3′ scRNA-seq method. Briefly, 20 μL of 1000 cells/μL PBMC suspension from each subject/condition were combined, and 33.8 μL of cell suspension (total cell number = 33,800) was mixed with 66.2 μL of RT reaction mix before being added to a chromium microfluidics chip already loaded with 40 μL of barcoded beads and 270 μL of partitioning oil. The chip was then placed within the chromium controller where single cells and barcoded beads were encapsulated together within oil droplets. Reverse transcription was then performed within the oil droplets to produce barcoded cDNA. Barcoded cDNA was isolated from the partitioning oil using Silane DynaBeads (Thermo Fisher Scientific, Dynabeads MyONE Silane, Cat# 37002D) before amplification by PCR. Cleanup and size selection was performed on amplified cDNA using SPRIselect beads (Beckman-Coulter, SPRIselect, Cat# B23317) and cDNA quality was assessed using an Agilent 2100 BioAnalyzer and the high-sensitivity DNA assay (Agilent, High-Sensitivity DNA Kit, Cat# 5067-4626). Sequencing libraries were generated from cleaned, amplified cDNA using the 10X Chromium Kit's including reagents for fragmentation, sequencing adaptor ligation, and sample index PCR. Between each of these steps, libraries were cleaned, and size selected using SPRIselect beads. Final quality of cDNA libraries was once again assessed using the Agilent BioAnalyzer High-Sensitivity DNA assay, and quality-confirmed libraries were sequenced using Illumina's NextSeq platform. Additional primers were included in the cDNA amplification step to amplify the TotalSeq oligonucleotide tags. During the post-amplification cleanup, supernatant containing amplified TotalSeq tags was collected and processed parallel to the standard 10X library fraction. All reagents are listed in Supplementary Table 2.

**PBMCs from healthy individuals used in in vitro assays**. Fifteen milliliters of fresh whole blood from healthy donors (Research Blood Components, Boston MA) were diluted 1:1 with warm PBS + 2% FBS, mixed, and gently layered atop 30 mL Ficoll-Paque density gradient medium (GE Healthcare, Ficoll-Paque PLUS, Cat# 17-1440) in 50 mL conical tubes. This process was repeated seven times in processing 100 mL of blood. Tubes were centrifuged for 20 min at 1200 × g to separate leukocytes from red blood cells and plasma. The leukocyte-containing buffy coat was carefully transferred into new tubes, washed with warm PBS + 2% FBS, counted, resuspended in DMSO, and aliquoted. Isolated cells were then stored in liquid nitrogen until later experimental use. Due to the anonymous nature of the procured whole-blood samples, no ethical oversight was required from the Boston University IRB for these samples. All reagents are listed in Supplementary Table 2.

**Controlled substances**. Solid morphine sulfate (Sigma Aldrich, Cat# M8777-25G) was obtained with approval and oversight from the controlled substances sub-office of the Boston University Department of Environmental Health and Safety. Aliquots of a 10 mM stock solution were prepared and stored for further use in experimentation. All reagents are listed in Supplementary Table 2

**Morphine titration in PBMCs from healthy individuals**. Normal PBMCs were revived in leukocyte-supporting complete RPMI medium and plated onto non-tissue culture-treated 96-well plates at a density of 2.0e5 cells/well (two wells per condition, 4.0e5 cells total). Cells were treated either with a mock treatment or a titration of morphine sulfate in RPMI complete medium (0, 10, 100 μM) for 24 h. At the end of the morphine incubation either medium or LPS (final concentration 100 ng/mL) was added to the wells and cells were incubated for further 3 h, at the end of which the cells were collected, washed, and processed for total RNA using the ZymoPure QuickRNA MiniPrep kit (Zymo Research, Cat# R1055). RNA samples were then used to perform RT-qPCR. All reagents are listed in Supplementary Table 2

**RT-qPCR analysis**. Total RNA was isolated from cells using the ZymoPure QuickRNA MiniPrep kit. cDNA was synthesized using ~50 ng of total RNA per sample (Thermo Fisher, SuperScript IV First-Strand Synthesis System, Cat# 18091200). Two microliters of cDNA per reaction was used to perform qPCR (Fisher Scientific, PowerUp SYBR Greßen Master Mix, Cat# A25742) with primers against transcripts of the Interferon target gene ISG15 (IDT; Supplementary Table 3) We used primers against ActB (IDT, primer sequences in supplement) as a housekeeping gene control. All reagents are listed in Supplementary Table 2.

**scRNA-seq of in vitro morphine treatment with healthy PBMCs**. Cells were treated either with a mock treatment or 100 μM of morphine sulfate in RPMI complete medium for 24 h. At the end of the morphine incubation either medium

or LPS (final concentration 100 ng/mL) was added to the wells and cells were incubated for additional 3 h, at the end of which the cells were collected and washed. After treatment, all cells were collected, and an equal number of cells per patient sample was subjected to cell hashing for scRNA-seq using 1 μg of TotalSeq antibodies (Supplementary Table 5). Hashtagged cells were then washed, diluted to 1000 cells/μL, and pooled before being used to perform the 10X Genomics Chromium Single Cell 3′ method. Briefly, 20 μL of 1000 cells/μL PBMC suspension from each subject/condition were combined, and 33.8 μL of cell suspension (total cell number = 33,800) was mixed with 66.2 μL of RT reaction mix before being added to a chromium microfluidics chip already loaded with 40 μL of barcoded beads and 270 μL of partitioning oil. The chip was then placed within the chromium controller where single cells and barcoded beads were encapsulated together within oil droplets. Reverse transcription was then performed within the oil droplets to produce barcoded cDNA. Barcoded cDNA was isolated from the partitioning oil using Silane DynaBeads (Thermo Fisher Scientific, Dynabeads MyONE Silane, Cat# 37002D) before amplification by PCR. Cleanup/size selection was performed on amplified cDNA using SPRIselect beads (Beckman-Coulter, SPRI-select, Cat# B23317) and cDNA quality was assessed using an Agilent 2100 BioAnalyzer and the High-Sensitivity DNA assay (Agilent, High-Sensitivity DNA Kit, Cat# 5067-4626). Sequencing libraries were generated from cleaned, amplified cDNA using the 10X Chromium Kit's including reagents for fragmentation, sequencing adaptor ligation, and sample index PCR. Between each of these steps, libraries were cleaned and size selected using SPRIselect beads. Final quality of cDNA libraries was once again assessed using the Agilent BioAnalyzer High-Sensitivity DNA assay, and quality-confirmed libraries were sequenced using Illumina's NextSeq platform. All reagents are listed in Supplementary Table 2.

**Single cell analysis**. LPS treatment: RNA-seq processing and downstream analysis: We used CellRanger version 2.1.0 (10X Genomics) to pool and process the raw RNA sequencing data. First, using CellRanger mkfastqc pipeline, each sample sequencing library was demultiplexed based on the sample index read to generate FASTQ files for the paired-end reads. STAR aligner[39] was used to align reads to the human reference genome (GRCh38) through the CellRanger count pipeline. After alignment, all sample libraries were equalized to the same sequencing depth (each sample cell is downsampled to have the same confidently mapped reads per cell) and aggregated together subsequently to generate a gene-cell barcode matrix using CellRanger aggr pipeline.

After data aggregation, we performed all filtering, normalization, and scaling of data using Seurat suite version 2.3 (refs. [12,13]). Cells with less than 300 and greater than 2000 detected genes were filtered out, as well as cells with greater than 10,000 UMIs and greater than 10% mitchondrial counts were filtered out. Genes that were detected in less than 10 cells were removed. Gene counts for each cell were normalized by total expression, multiplied by a scale factor of 10,000 and transformed to log scale.

PCA based on the highly variable genes detected (dispersion of 2) was performed for dimension reduction and the top 20 principal components (PCs) were selected. We clustered cells based on graph-based methods (KNN and Louvain community detection method) implemented in Seurat. The clusters and other known annotations were visualized using t-stochastic neighbor embedding (t-SNE)[40].

Cell hashing processing and analysis: For hashtag oligo (HTO) quantification, we first ran Cite-seq-Count[15,41] on the HTO fastq files to process the HTO reads with the parameters specific to 10X Genomics single-cell 3′ v2 data as stated in https://github.com/Hoohm/CITE-seq-Count. In addition, we used CellRanger v.3.0.2 (10X Genomics) to process the raw sequencing RNA reads and Seurat suite for downstream analyses. To identify the cells sample-of-origin, we demultiplexed the HTOs and removed doublets and ambiguous cells using the Seurat pipeline for demultiplexing as mentioned in https://satijalab.org/seurat/hashing_vignette.html.

IFNβ treatment: After demultiplexing the HTOs, we performed all downstream analyses as described above. Cells with less than 200 and greater than 2500 detected genes were filtered out, as well as cells with greater than 5% mitochondrial counts were filtered out.

In vitro morphine treatment: After demultiplexing the HTOs, we performed all downstream analyses as described above. Cells with less than 500 and greater than 3000 detected genes were filtered out, as well as cells with greater than 5% mitochondrial counts were filtered out.

RNA sequencing DE analysis: To identify peripheral immune subpopulations, we performed differential expression analysis using Wilcoxon rank-sum test between clusters to identify top expressing genes for each cluster for cell type identification implemented in Seurat. Cell-type-specific gene signatures were determined from the overlap of more highly expressed and canonical gene markers.

We performed differential expression analysis for each cell type between control cells and opioid-dependent cells using Model-based Analysis of Single Cell Transcriptomics (MAST)[42]. Utilizing this method, we fit a hurdle model modeling condition and the centered cellular detection rate (cngeneson), and then performed a likelihood ratio test dropping the condition term to identify genes upregulated and downregulated in opioid-dependent samples compared to controls. Differentially expressed genes were evaluated according to their log fold change (greater than log2(1.5)) and adjusted p values (0.05). All figures generated using ggplot2 R package[43].

Enrichment analysis: We performed gene enrichment analysis of the list of differential genes between opioid-dependent individuals and non-dependent controls for each cell type using Metascape[44] online tool (http://metascape.org/). The enrichment analysis was run using default settings, and was assessed and visualized through a heatmap of significance ($-\log(p\ value)$). All heatmaps were generated using ComplexHeatmap R package and color scale generated using dependent R package circlize[45].

**Reporting summary**. Further information on research design is available in the Nature Research Reporting Summary linked to this article.

## Data availability

Processed cell-hashing scRNA-seq data are available from GEO under accession GSE128879. Raw scRNA-seq data of opioid-dependent individuals and non-dependent controls are available upon request from dbGaP under accession phs000277.v2 at [https://www.ncbi.nlm.nih.gov/projects/gap/cgi-bin/study.cgi?study_id=phs000277.v2.p1]. We have included all processed scRNA-seq datasets on Single Cell Portal, including the cell barcodes, t-SNE coordinates, and other available characteristics (sample ID, cell type, age, sex, and comorbidities). The processed scRNA-seq data of LPS-treated PBMCs from opioid-dependent individuals and controls are available at [https://singlecell.broadinstitute.org/single_cell/study/SCP587/]. The processed scRNA-seq data of IFNβ-treated PBMCs from opioid-dependent individuals and controls are available at [https://singlecell.broadinstitute.org/single_cell/study/SCP589/]. The processed scRNA-seq data of in vitro morphine-treated PBMCs are available at [https://singlecell.broadinstitute.org/single_cell/study/SCP591/]. The source data underlying Figs. 1–d, 3a, b, and Supplementary Figs. 4–5, 13, 22 are provided in the Source Data file.

## Code availability

The original R scripts for Seurat processing and cell hashing are available on github [https://github.com/satijalab/seurat]. All custom-made code to reproduce the analyses and figures reported in this paper are available on github (https://github.com/tanya-karagiannis/scRNAseq-PBMC-opiate).

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

## Acknowledgements

We thank Dr. Todd A. Blute for his technical support on flow cytometry analysis. We would like to thank Dr. David J. Waxman for critical reading of the manuscript. The work was supported by NIH (R61DA047032 to C.S.C.). C.S.C. was also supported by Boston University Data Science Faculty Fellowship. T.T.K was also supported by NIH institutional training grant T32GM100842. CATS data collection was funded by R01DA17305 and E.C.N. is supported by R01DA042620, R01DA046436, and R33DA041883. A.J.H. is supported by NIH R61DA047032.

## Author contributions

T.T.K. designed and performed the analysis, interpreted the results, and wrote the paper. J.P.CJr designed and performed the experiments, interpreted the results, and wrote the paper. B.G. helped with the experiments. A.J.H. helped to interpret the results and helped edit the paper. N.G.M. provided PBMC samples from opioid-dependent individuals and non-dependent controls and helped edit the paper. M.Y. helped design statistical tests for scRNA-seq analysis. E.C.N. provided PBMC samples from opioid-dependent individuals and non-dependent controls, helped put together information on patient medical history, and helped edit the paper. C.S.C. conceived and oversaw the project, provided guidance, interpreted the results and wrote the paper.

## Competing interests

The authors declare no competing interests.
