## [Peer Review File · Nature Communications]

Reviewers' comments:

Reviewer #1, expertise in inflammation impact on opioid receptors (Remarks to the Author):

Gaining a better understanding of the effects that opioids have on the immune system is timely and of particular interest given the ongoing "opioid epidemic". This study utilizes an innovative tool, single cell RNA sequencing (sc-RNA-seq) to characterize the effects of opioid usage on inflammatory pathways in human PBMC.

Likely, the most impactful aspect of this report is the use of sc-RNA-seq and the general finding that antiviral genes in PBMC are particularly susceptible to chronic opioid usage. The supportive findings from the in vitro exposure to morphine are certainly a strength.

However, there are substantial concerns with the manuscript in the present form. This manuscript should be substantially improved by addressing the following concerns:

Major concerns:

Figure legends- Details on the statistical tests used and specific comparisons made should be detailed in the figure legends

Discussion- there are numerous reports describing the immunomodulatory effects of opioids, yet there is very little actual discussion in the context of the literature.

Please state specifically how these findings are novel and the extent to which they advance the field, both in terms of the methods and opioid effects on immune function.

Also, there is evidence in the literature that opioids may impact inflammatory signaling through receptors other than opioid receptors. See for example LR Watkins et al., and actions at TLR4; or the apparent MOR-independent anti-inflammatory actions of beta-funaltrexamine reported by RL Davis et al.

Pg. 3, paragraph 2: regarding data in Fig. 3 the authors state there was inhibition "as quickly as 3h after morphine treatment"... This is confusing, as it seems the cells were treated for 24h then 3 h of LPS treatment. Please clarify.

Other concerns;

Abbreviations for opioid receptor subtypes- recommend MOR, DOR, KOR rather than MOP, DOP, and KOP

The summary paragraph should include more information/insights related to opioid effects on immune signaling and health consequences of opioid abuse.

Consider a more basic description of sc-RNA-seq in the introduction that would benefit readers with less experience in the field

Reviewer #2, expert in TLR signaling (Remarks to the Author):

Summary:

This paper asks how chronic opioid use affects the peripheral immune system, a question that has

been fairly controversial, with previous studies being inconclusive. To more definitively and comprehensively answer this question, the authors conduct single cell RNA seq on PBMC from opioid dependent (mostly former heroin users on opioid replacement) and neighborhood controls; or on PBMC stimulated ex vivo with LPS. In addition, they perform in vitro experiments, exposing PBMC from healthy volunteers to morphine, followed by LPS. In each case, they find that opioid exposure (in vivo or in vitro) reduces the expression of the antiviral gene program in several immune cell populations.

This is an important finding that would be of interest to a multidisciplinary audience, including immunologists, pain biologists, those studying neuro-immune interactions, and clinicians treating patients with opioids or patients with opioid dependence.

Concerns:

The main finding is strong, however, there are a few concerns listed below that could be more explicitly addressed in the manuscript text.

Major:

1. The mechanism the authors propose for inhibition of interferon stimulated gene pathway is confusing and not supported by the data--should explain it more clearly (perhaps with a supplementary model). Monocytes are the main cells that express TLR4 and respond to LPS, while lymphocytes usually respond to products (IFN and TNF, etc) made by monocytes to activate an antiviral response (in the context of LPS). Upon the ex vivo stimulation the monocytes seem to recover their ability to make ISG (interferon stimulated genes), so would be expected to be secreting paracrine factors to activate the lymphocytes, however the lymphocyte gene expression is still decreased. The authors propose that there is (somehow) an upregulation of negative regulators of interferon signaling by opioids in the naive state, that then 1) inhibits antiviral gene expression in naive monocytes and 2) in lymphocytes and NK cells after LPS stimulation (perhaps preventing them from responding to the ifn from monocytes?? although the authors do not explicitly suggest this). This is shown in Figure S11, which is really not convincing for differential expression of negative regulators. perhaps stronger evidence could be obtained by rtqpcr for multiple neg regulators from the in vitro experiments...would also be great to try (at least in vitro) a TLR7 ligand that would stimulate interferon response directly in lymphocytes...

2. The authors consistently refer to "three major innate immune modules" described by Shalek et al. While this may be a useful construct for this current study, as it separates out the antiviral program from other modules, it needs to be made clear that 1) this is ONE way of making gene expression groupings based mostly on timing of expression; 2) the Shalek set of experiments was performed in mouse bone marrow derived dendritic cell cultures, and therefore it is really unclear whether/how it applies to the multiple human peripheral blood cell types studied here, which may, and very likely do have different temporal patterns of gene expression. For example, are the "peak" inflammatory genes described in the Regev paper uniformly "peak" in CD4 lymphocytes? that question can't be answered by this present study which examines only a single timepoint. So I think the "the major innate immune modules" needs to be reworded/toned down/qualified.

Minor:

1. Would be nice to have a better demographic description of the "opioid dependent" populations studied here. For example, although in the discussion/methods it's implied they were mostly injection heroin users who are now on opioid maintenance, I didn't see where details of comorbidities (i.e. chronic pain, medical illness, viral infection, which would definitely affect interferon response) were provided. Perhaps these could be listed in a supplementary table.

2. throughout the paper consider changing the nomenclature to LPS treatment: monocytes, LPS treatment: activated T cells, etc... Reading the heading "LPS activated T cells" is confusing, as it implies that LPS was applied to T cells and activated them.

3. would be nice to add a short explanation of why opioid receptor expression couldn't be detected in the present study and why the authors had to refer to a previous study. Similarly for ifnr, tnfr, etc...

Reviewer #3, expert in scRNA-seq (Remarks to the Author):

The authors describe a study about the effect of chronic opioid usage on the peripheral immune system using single-cell transcriptomics analysis. This study reveals a widespread suppression of interferon-stimulated and antiviral signatures in multiple immune cell types in the opioid dependent individuals. Especially, both ex vivo and in vitro experiments using LPS stimulation show the same suppression of antiviral gene signatures, which suggests the adverse effect from opioid usage on the defense response and immune system. I think this work is interesting and some findings about the "antiviral gene activation" are important. However, I have some concerns on the computational side and also validation experiments.

1. For the qc step of the single-cell data, the authors use filters like number of genes detected. Did you also filter out bad quality cells based on percent of mitochondrial genes detected? Mitochondrial levels reflect the variability of cell damage, and high mitochondrial content suggests cell damage. Fig 1C shows that mitochondrial genes are involved in the downstream analysis and upregulated in the non-opioid controls. Also, please annotate which side (left or right) on the x-axis is control/opioid in Fig 1C.

2. For the single-cell analysis, the authors performed a normalization step based on sequencing depth when aggregating all samples into one matrix. Can you specify details about how you normalized the data? Did you perform scaling before or after merging all samples? In my experience, it is best to perform such operations on the merged dataset, because you don't want to lose any cell types that are unique to one dataset.

3. Single-cell transcriptomic is powerful, but it is easy to find technical/batch effects. For the single-cell integrative analysis, it looks like the authors did not correct sample by sample batch effect before clustering. There are some clear sample-specific batch effect sub-clusters in the clustering analysis (Figure 1B right). It is unclear if there are any "non-opioid controls vs opioid" differences or sample-specific differences. Please correct effect across samples before performing clustering. Multiple single-cell RNA-seq batch effect correction methods are already available: Harmony (<https://github.com/immunogenomics/harmony>); BBKNN (<https://github.com/Teichlab/bbknn>), or multiCCA related tools.

4. Figure 2B and Figure S6-S10 are very confusing. The authors are trying to say "we found widespread suppression of antiviral genes in opioid dependent cells across LPS-stimulated immune subpopulations", which is a key observation of this study. Two questions: First, the heatmap is not very informative (Figure 2B), because it is hard to tell if the difference between control and opioid for the LPS immune cells are significant. Also, can you label the key antiviral genes in the heatmap (for example ISG15 and TLR4)? Second, does the violin plot show the averaged gene expression for all the genes from the left heatmap or just one representative gene? Take Fig S10B left subpanel as an example, I am confused about what the exact comparison is for the "*****" in the analysis. Do you mean average of "C1-C7" comparing to average of "O1-O7"? For this kind of analysis, I also recommend building some mixed linear model to set LPS/non-LPS and donor as fix/random effects, and then test and report the most differential genes and corresponding statistics between control and opioid groups.

5. For the visualization purpose, can you plot cells as dots on top of the violin plots from Fig 2 and Fig

3?

6. For gene enrichment analysis, can you use MSigDB hallmark gene sets? The MSigDB was designed and collected based on immunological functional experiments.

7. I am wondering why there were no dendritic cells in the opioid PBMC. Dendritic cells might be involved in the interaction or co-expression with T cells.

8. In Fig 1D, why didn't the CD8 LPS cluster respond to virus or related with interferon signaling? Maybe the authors only use type I interferon IFNalpha and IFNbeta LPS stimulation instead of IFNgamma? However, in the cell hashing experiment (Figure 3C), the authors observed a suppression of core antiviral genes in response to morphine exposure for the CD8+ T cells. Can you explain the inconsistency here?

9. In the manuscript, "We have observed strong suppression of the antiviral gene program in response to LPS in PBMCs of opioid dependent individuals, and very modest suppression of the inflammatory modules". I see that there is no significant suppression of inflammatory modules in response to LPS of opioid individuals. However, as an indirect effect of opioids, morphine may cause suppression of inflammatory cytokines. Can you show the expression pattern of several typical inflammatory or pro-inflammatory cytokines, for example IL1B?

10. As a validation, can you validate these findings of "strong suppression of the antiviral gene program" at the protein level using maybe either 1) cite-seq for both mRNA and protein detection for each individual cell, or 2) antiviral protein staining using microscope imaging? Most of the current results only rely on transcriptomic analysis. It is important to know if there are any suppression of antiviral and interferon proteins from the opioid dependent individuals.

11. Data publication: Looks like the raw single-cell data of "Accession "GSE128879" is currently private and is scheduled to be released on Mar 25, 2022". Please make sure it is available to be downloaded.

12. Can you create a website or web portal so that readers can browser the data and results? It will also be a good resource for the folks in this community.

Reviewers' comments:

Reviewer #1, expertise in inflammation impact on opioid receptors (Remarks to the Author):

Gaining a better understanding of the effects that opioids have on the immune system is timely and of particular interest given the ongoing “opioid epidemic”. This study utilizes an innovative tool, single cell RNA sequencing (sc-RNA-seq) to characterize the effects of opioid usage on inflammatory pathways in human PBMC.

Likely, the most impactful aspect of this report is the use of sc-RNA-seq and the general finding that antiviral genes in PBMC are particularly susceptible to chronic opioid usage. The supportive findings from the in vitro exposure to morphine are certainly a strength.

=> We would like to thank the Reviewer for their positive comments and support of our manuscript.

However, there are substantial concerns with the manuscript in the present form. This manuscript should be substantially improved by addressing the following concerns:

Major concerns:

1. Figure legends- Details on the statistical tests used and specific comparisons made should be detailed in the figure legends

=> We regret any of the missing information and have included more details in the legends of Figure 2-3 as well as Supplementary Figures 6-12, 15, 17-S21.

2. Discussion- there are numerous reports describing the immunomodulatory effects of opioids, yet there is very little actual discussion in the context of the literature.

=> We fully agree with the Reviewer, and thank the Reviewer for the suggestion. We have added a new section of discussion to describe current literature about immunomodulatory effects of opioids (page 4-5).

3. Please state specifically how these findings are novel and the extent to which they advance the field, both in terms of the methods and opioid effects on immune function.

=> We thank the Reviewer for the suggestion. We have added a new section of discussion to state specifically how our findings are novel both in terms of the methods and opioid effects on immune function (page 5).

4. Also, there is evidence in the literature that opioids may impact inflammatory signaling through receptors other than opioid receptors. See for example LR Watkins et al., and actions at TLR4; or the apparent MOR-independent anti-inflammatory actions of beta-funaltrexamine reported by RL Davis et al.

=> We thank the Reviewer for the suggestion. In the same paragraph in the discussion (addressing point 2), we have added discussion about these MOR-independent effects, especially in the CNS and have cited these two papers (page 5).

5. Pg. 3, paragraph 2: regarding data in Fig. 3 the authors state there was inhibition “as quickly as 3h after morphine treatment”... This is confusing, as it seems the cells were treated for 24h then 3 h of LPS treatment. Please clarify.

=> We thank the Reviewer for pointing this out. We performed two in vitro morphine treatment experiments followed by RT-qPCR. For the first experiment, PBMCs were pretreated with morphine for 24 hours followed by 3 hours of LPS treatment (Figure 3a). For the second experiment, PBMCs were pretreated with morphine for 3 hours followed by 3 hours of LPS treatment (Figure 3b). We have updated the text to clarify that the statement of inhibition “as quickly as 3h after morphine treatment” is in reference to the second RT-qPCR experiment (page 4). ORIGINAL: Furthermore, this inhibition was observed as quickly as three hours after morphine pretreatment

UPDATED: Furthermore, this inhibition was also detectable after only three hours of morphine pretreatment followed by three hours of LPS treatment

Other concerns;

1. Abbreviations for opioid receptor subtypes- recommend MOR, DOR, KOR rather than MOP, DOP, and KOP

=> We thank the Reviewer for their recommendation. We have changed the abbreviations for the opioid receptor subtypes to the recommended abbreviations in the main text (page 5) and in Supplementary Figure 22.

2. The summary paragraph should include more information/insights related to opioid effects on immune signaling and health consequences of opioid abuse.

=> We thank the Reviewer for their suggestion. We have added a summary paragraph in the discussion section that includes discussion on the effect of opioids on immune signaling and health consequences (page 6).

3. Consider a more basic description of sc-RNA-seq in the introduction that would benefit readers with less experience in the field

=> We thank the Reviewer for this helpful suggestion. We have added a new paragraph in the introduction section to provide a more basic introduction of scRNA-seq.

Reviewer #2, expert in TLR signaling (Remarks to the Author):

Summary:

This paper asks how chronic opioid use affects the peripheral immune system, a question that has been fairly controversial, with previous studies being inconclusive. To more definitively and comprehensively answer this question, the authors conduct single cell RNA seq on PBMC from opioid dependent (mostly former heroin users on opioid replacement) and neighborhood controls; or on PBMC stimulated ex vivo with LPS. In addition, they perform in vitro experiments, exposing PBMC from healthy volunteers to morphine, followed by LPS. In each case, they find that opioid exposure (in vivo or in vitro) reduces the expression of the antiviral gene program in several immune cell populations.

This is an important finding that would be of interest to a multidisciplinary audience, including immunologists, pain biologists, those studying neuro-immune interactions, and clinicians treating patients with opioids or patients with opioid dependence.

=> We thank the Reviewer for their enthusiasm for our work.

Concerns:

The main finding is strong, however, there are a few concerns listed below that could be more explicitly addressed in the manuscript text.

Major:

1. The mechanism the authors propose for inhibition of interferon stimulated gene pathway is confusing and not supported by the data--should explain it more clearly

(perhaps with a supplementary model). Monocytes are the main cells that express TLR4 and respond to LPS, while lymphocytes usually respond to products (IFN and TNF, etc) made by monocytes to activate an antiviral response (in the context of LPS). Upon the ex vivo stimulation the monocytes seem to recover their ability to make ISG (interferon stimulated genes), so would be expected to be secreting paracrine factors to activate the lymphocytes, however the lymphocyte gene expression is still decreased. The authors propose that there is (somehow) an upregulation of negative regulators of interferon signaling by opioids in the naive state, that then 1) inhibits antiviral gene expression in naive monocytes and 2) in lymphocytes and NK cells after LPS stimulation (perhaps preventing them from responding to the ifn from monocytes?? although the authors do not explicitly suggest this). This is shown in Figure S11, which is really not convincing for differential expression of negative regulators. perhaps stronger evidence could be obtained by rtqpcr for multiple neg regulators from the in vitro experiments...would also be great to try (at least in vitro) a TLR7 ligand that would stimulate interferon response directly in lymphocytes...

=> We fully agree with the Reviewer and thank the Reviewer for the suggestion. We agree with the Reviewer that the differential expression of interferon regulators (original Figure S11) is not very convincing and have removed this figure. We take the Reviewer's suggestion in using another stimulus that will activate interferon response in lymphocytes. We looked at the expression of TLR7 receptor in population level RNA-seq data from PBMCs of healthy individuals from a previous study (Corces et al., Nature Genetics 2016). We found that TLR7 is only expressed in B cells and monocytes in PBMCs. We then checked IFNAR1 and IFNAR2 expression and found both of them are expressed at high levels in each of the immune cell subpopulations in PBMCs (CD4+ T cells, CD8+ T cells, B cells, NK cells and monocytes) (Supplementary Figure 13). We decided to stimulate PBMCs from opioid-dependent patients vs controls with IFN β for 3 hours and followed by scRNA-seq profiling. We found that IFN β treatment did not produce the antiviral gene program suppression phenotype that we observed when PBMCs from opioid users were treated with LPS. Because IFN β activates the antiviral gene program directly through the IFNAR receptors, we speculate the modulatory effect of opioids are affecting a component in the the TLR4/TRIF signalling cascade, which could explain the observed suppression of the antiviral gene program in response to LPS (Figure 1 and 2). As the Reviewer has pointed out, the in vivo experiment with opioid-dependent individuals demonstrated suppression of antiviral genes in naive monocytes and upon LPS treatment in monocytes and also other cell types, while the in vitro experiment shows that antiviral genes were slightly suppressed after 24 hours of morphine treatment in monocytes. The induction of antiviral genes upon LPS

treatment were not suppressed in morphine treated monocytes but were mildly suppressed in other cell types such CD4+ T cells and NK cells. We speculate that for the in vitro case, the dysregulation of the TLR4/TRIF component happens in the basal condition upon 24 hours of pretreatment with morphine without LPS treatment in monocytes and then this affected the basal expression of antiviral genes in other cell types such as CD4+ T cells, which might have caused the mild suppression of antiviral genes upon LPS treatment. We fail to find an explanation for the lack of suppression of antiviral genes upon LPS treatment in the monocytes in the in vitro experiment. We would like to suggest that understanding of the mechanism beyond what we have observed using scRNA-seq is outside of the scope of the current study.

We have included a paragraph in the Results section to describe this new scRNA-seq dataset from IFN β treatment (Page 3-4) and have added a paragraph to discuss the IFN β result and our speculation about the effect of opioid in dysregulating a component in the TLR4/TRIF pathway (Supplementary Figure 14-15).

2. The authors consistently refer to "three major innate immune modules" described by Shalek et al. While this may be a useful construct for this current study, as it separates out the antiviral program from other modules, it needs to be made clear that 1) this is ONE way of making gene expression groupings based mostly on timing of expression; 2) the Shalek set of experiments was performed in mouse bone marrow derived dendritic cell cultures, and therefore it is really unclear whether/how it applies to the multiple human peripheral blood cell types studied here, which may, and very likely do have different temporal patterns of gene expression. For example, are the "peak" inflammatory genes described in the Regev paper uniformly "peak" in CD4 lymphocytes? that question can't be answered by this present study which examines only a single timepoint. So I think the "the major innate immune modules" needs to be reworded/toned down/qualified.

=> We thank the Reviewer for this comment. We have toned down the "three major innate immune modules" and have made a statement to clarify that this is one way of classifying innate immune response gene expression programs (page 3).

Minor:

1. Would be nice to have a better demographic description of the "opioid dependent" populations studied here. For example, although in the discussion/methods it's implied they were mostly injection heroin users who are now on opioid maintenance, I didn't see

where details of comorbidities (i.e. chronic pain, medical illness, viral infection, which would definitely affect interferon response) were provided. Perhaps these could be listed in a supplementary table.

=> We thank the Reviewer for the suggestion and we have updated Supplementary Table 5 with details of known comorbidities.

2. throughout the paper consider changing the nomenclature to LPS treatment: monocytes, LPS treatment: activated T cells, etc... Reading the heading "LPS activated T cells" is confusing, as it implies that LPS was applied to T cells and activated them.

=> We thank the Reviewer for this suggestion as it removes any confusion between the experimental details and cell types identified. We have made changes in the main text and in figure legends such as "LPS treated populations: NK cells, ..." to help clarify.

3. would be nice to add a short explanation of why opioid receptor expression couldn't be detected in the present study and why the authors had to refer to a previous study. Similarly for ifnr, tnfr, etc...

=> We thank the Reviewer for their recommendation. We have made a statement in the discussion (page 5) to explain this: "Due to the nature of single cell assays such as scRNA-seq, the expression level of opioid receptors at a single cell level was very low or not detectable."

Reviewer #3, expert in scRNA-seq (Remarks to the Author):

The authors describe a study about the effect of chronic opioid usage on the peripheral immune system using single-cell transcriptomics analysis. This study reveals a widespread suppression of interferon-stimulated and antiviral signatures in multiple immune cell types in the opioid dependent individuals. Especially, both ex vivo and in vitro experiments using LPS stimulation show the same suppression of antiviral gene signatures, which suggests the adverse effect from opioid usage on the defense response and immune system. I think this work is interesting and some findings about the "antiviral gene activation" are important. However, I have some concerns on the computational side and also validation experiments.

=> We thank the Reviewer for their insightful feedback and thoughtful advice. We also thank the Reviewer for their support of our findings.

1. For the qc step of the single-cell data, the authors use filters like number of genes detected. Did you also filter out bad quality cells based on percent of mitochondrial genes detected? Mitochondrial levels reflect the variability of cell damage, and high mitochondrial content suggests cell damage. Fig 1C shows that mitochondrial genes are involved in the downstream analysis and upregulated in the non-opioid controls. Also, please annotate which side (left or right) on the x-axis is control/opioid in Fig 1C.

=> We thank the Reviewer for this suggestion. We filtered our data based on percent of mitochondrial genes detected > 10% and we have made additions in the Methods section describing this qc step (page 11). In addition, we have annotated Fig 1C as recommended to clarify which side is upregulated in control and in opioid.

2. For the single-cell analysis, the authors performed a normalization step based on sequencing depth when aggregating all samples into one matrix. Can you specify details about how you normalized the data? Did you perform scaling before or after merging all samples? In my experience, it is best to perform such operations on the merged dataset, because you don't want to lose any cell types that are unique to one dataset.

=> We thank the Reviewer for their comment. After alignment, we performed a normalization step to equalize read depth between sample libraries using CellRanger aggr pipeline where each sample cell is downsampled to have the same confidently mapped reads per cell. Using the same pipeline, the sample libraries were aggregated subsequently to generate a pooled gene-cell barcode matrix. After data aggregation, we performed scaling of the data. We have added these details in the Methods section (page 10) to describe the order of steps in processing and analysis of the single cell data.

3. Single-cell transcriptomic is powerful, but it is easy to find technical/batch effects. For the single-cell integrative analysis, it looks like the authors did not correct sample by sample batch effect before clustering. There are some clear sample-specific batch effect sub-clusters in the clustering analysis (Figure 1B right). It is unclear if there are any "non-opioid controls vs opioid" differences or sample-specific differences. Please correct effect across samples before performing clustering. Multiple single-cell RNA-seq batch effect correction methods are already available: Harmony (<https://github.com/immunogenomics/harmony>); BBKNN (<https://github.com/Teichlab/bbknn>), or multiCCA related tools.

=> In the experimental design of this study, for the patient samples, we ran all non-dependent controls and opioid-dependent samples with the same treatment condition in the same experimental batch with 10X (two batches, one for naive PBMCs and another for LPS treated PBMCs). We only compare between the two patient categories within each of the treatment conditions (naive or LPS treated). For in vitro experiments, we performed scRNA-seq on samples for each treatment condition within the same 10X well by using hashtag (one well for LPS treatment experiment, another well for IFN β treatment experiment). Therefore, we do not need to account for technical/batch effects in this study.

4. Figure 2B and Figure S6-S10 are very confusing. The authors are trying to say “we found widespread suppression of antiviral genes in opioid dependent cells across LPS-stimulated immune subpopulations”, which is a key observation of this study. Two questions: First, the heatmap is not very informative (Figure 2B), because it is hard to tell if the difference between control and opioid for the LPS immune cells are significant. Also, can you label the key antiviral genes in the heatmap (for example ISG15 and TLR4)? Second, does the violin plot show the averaged gene expression for all the genes from the left heatmap or just one representative gene? Take Fig S10B left subpanel as an example, I am confused about what the exact comparison is for the “****” in the analysis. Do you mean average of “C1-C7” comparing to average of “O1-O7”? For this kind of analysis, I also recommend building some mixed linear model to set LPS/non-LPS and donor as fix/random effects, and then test and report the most differential genes and corresponding statistics between control and opioid groups.

=> We thank the Reviewer for these comments and recommendations. To help inform the heatmaps such as those in Figure 2B, we labelled key antiviral and inflammatory genes in all heatmaps. The violin plots shown in Figure 2B represent the average expression of all genes in a geneset for each cell. The statistical comparison in the violin plots are between the average geneset expression of all samples in controls (C1-C7) and the average geneset expression of all samples in opioid (O1-O7). We included more details in the figure and figure legends of Figure 2B, 3B and Supplementary Figures 6-12, 15, 17-21 to make this clear.

As recommended by the Reviewer, we tried a generalized linear mixed model (GLMM) through MAST treating each sample as a random effect when we perform differential expression analysis. At the same time, we also compared the GLMM with a generalized linear model (GLM) without the random effect since we were concerned about the estimability of the random effect due to the small number of

samples that we had in this particular study. The ANOVA result that we got suggested that for the majority of genes, there is no significant difference between these two models. Since, MAST uses a discrete component and continuous component to inform the hurdle model, we did this on both components. We show the p-values for the ANOVA results for each gene for two cell types, naive monocytes (7 control and 7 opioid samples) and LPS treated: CD4+ T cells (3 control and 3 opioid samples) (Figure R1, R2). Based on this result, we have concluded that although the suggestion made by the Reviewer is methodologically reasonable, our results do not show a strong support for the inclusion of the random effect, which is expected given the small number of samples involved.

In addition, we do not set LPS/non-LPS as a factor in this analysis since the stimulation states clustered into separate clusters for each cell type (Figure 1B) and our analysis is focused for a specific cluster/population (Figure 1C).

Figure R1. Significance of the sample variable as a random effect in the model fit in MAST for differential expression analysis for naive monocytes.

Figure R2. Significance of the sample variable as a random effect in the model fit in MAST for differential expression analysis for LPS treated: CD4+ T cells (LPS CD4+ T cells).

5. (For the visualization purpose, can you plot cells as dots on top of the violin plots from Fig 2 and Fig 3?)

=> We thank the Reviewer for this suggestion. We have added cells as dots on the violin plots for Figures 2 and 3 as well as for Supplementary Figures 6-12, 15, 17-21.

6. For gene enrichment analysis, can you use MSigDB hallmark gene sets? The MSigDB was designed and collected based on immunological functional experiments.

=> We thank the Reviewer for this comment. In order to look into the enrichment of MSigDB hallmark gene sets, we had to perform this enrichment analysis with a different tool. The enrichment analysis for each cell type subpopulation was run using GSEA Prerank with MSigDB Hallmark gene sets and a ranked list of differential genes based on log fold change. The figure below shows significant normalized enrichment scores (NES) across all subpopulations for top 3 hallmark genesets: interferon gamma signalling (IFN γ Response), interferon alpha signalling (IFN α Response), and inflammatory response (Inflammatory) (Figure R3). The results show interferon signalling to be significantly enriched in controls for most LPS treated cell types and in naive monocytes, which we also observe in our original enrichment analysis with Metascape (Figure 1D). The inflammatory response geneset is shown to be significantly enriched in three cell types. However, we found several interferon response genes in this inflammatory geneset such as IFIT1, IRF1, and IFIT2. Since this inflammatory geneset contains a mixed set of genes that includes not only inflammatory genes but also interferon response genes, we decided to still use our original enrichment analysis with Metascape (Figure 1D).

Figure R3. Gene enrichment analysis results across PBMC subpopulations of control and opioid-dependent individuals. The enrichment analysis was performed for each cell type comparing enrichment in controls vs opioid across hallmark gene sets. The analysis is evaluated using normalized enrichment score (NES) with a negative score representing enrichment in controls (shown in green) and a positive score representing enrichment in opioid (shown in purple). White represents an insignificant result for the specific geneset.

7. I am wondering why there were no dendritic cells in the opioid PBMC. Dendritic cells might be involved in the interaction or co-expression with T cells.

=> We thank the Reviewer for this comment. We looked into marker genes for dendritic cells such as FCER1A, HLAD0A1, GPR183, and did not find a cluster that we could conclude to be dendritic cells. In human PBMC samples, dendritic cells make up a rare population, approximately 1-2% (<https://www.ncbi.nlm.nih.gov/books/NBK500157/>). With the capture rate of

scRNA-seq droplet-based methods, it is likely we missed this population or that the number of cells from dendritic cells are too small to form its own cluster.

8. In Fig 1D, why didn't the CD8 LPS cluster respond to virus or related with interferon signaling? Maybe the authors only use type I interferon IFNalpha and IFNbeta LPS stimulation instead of IFNgamma? However, in the cell hashing experiment (Figure 3C), the authors observed a suppression of core antiviral genes in response to morphine exposure for the CD8+ T cells. Can you explain the inconsistency here?

=> We thank the Reviewer for this point. We do observe a lower expression of core antiviral genes such as ISG15 and MX1 at a lower log fold change in opioid-dependent cells in the CD8 LPS cluster. Differentially expressed genes were evaluated according to their log fold change (greater than $\log_2(1.5)$) and adjusted p-values (0.05). These antiviral genes were below this cutoff for CD8 LPS cluster and therefore were not considered in the enrichment analysis shown in Fig 1D. To make this clear, we labelled these antiviral genes in our differential expression analysis results in Supplementary Figure 5 and state this observation in the main text (Page 3).

9. In the manuscript, "We have observed strong suppression of the antiviral gene program in response to LPS in PBMCs of opioid dependent individuals, and very modest suppression of the inflammatory modules". I see that there is no significant suppression of inflammatory modules in response to LPS of opioid individuals. However, as an indirect effect of opioids, morphine may cause suppression of inflammatory cytokines. Can you show the expression pattern of several typical inflammatory or pro-inflammatory cytokines, for example IL1B?

=> We thank the Reviewer for this comment and we fully agree with the Reviewer's suggestion on the inflammatory genes. We removed the sentence about the modest suppression of the inflammatory modules (page 6). We show the expression of several inflammatory genes in Supplementary Figure 7. We didn't include IL1B because the expression of IL1B was in general very low in all cell types.

10. As a validation, can you validate these findings of "strong suppression of the antiviral gene program" at the protein level using maybe either 1) cite-seq for both mRNA and protein detection for each individual cell, or 2) antiviral protein staining using microscope imaging? Most of the current results only rely on transcriptomic analysis. It is important to know if there are any suppression of antiviral and interferon proteins from the opioid dependent individuals.

=> We thank the Reviewer for this comment. Because current published CITE-seq protocol can only be used with cell surface proteins, the antiviral genes are mostly intracellular proteins. Thus we decided to try the Reviewer's second suggestion and utilize immunohistochemistry and microscope imaging to look at protein level of these antiviral genes. We performed immunofluorescence experiments using six patient samples (three opioid-dependent versus three opioid-free) from the same cohort as those samples used in our single cell transcriptomic experiments. We used antibodies against two of our most differentially expressed antiviral transcripts (ISG15 and MX1) to measure protein expression before and after treatment with LPS (3 hours). Protein expression was assessed by measuring the fluorescence intensity per cell using the cell profiler software suite (<https://cellprofiler.org/>). We measured these intensities in four imaging fields per sample/condition. However, using this methodology, we only observed a very modest induction of ISG15 and MX1 protein expression upon LPS treatment (only at most ~1.2 fold induction) (Figure R3). It is our belief that the sensitivity of this assay is not sufficient to detect the full extent of the protein level increase for ISG15 and MX1 upon LPS treatment. Despite this shortcoming, we observed a very modest suppression of ISG15 and MX1 protein level upon LPS treatment in opioid-dependent samples (Figure R4). The sensitivity of this assay is not sufficient to detect the full extent of the antiviral gene suppression suggested by our transcriptomic data, we decided not to include this data in the main text. We do, however, believe this data to be suggestive of the fact that our transcriptomic data are borne out at the protein level, and therefore feel comfortable sharing them with the Reviewer.

Figure R4. LPS response for ISG15 and MX1 in bulk PBMC as measured by immunofluorescence.

11. Data publication: Looks like the raw single-cell data of “Accession "GSE128879" is currently private and is scheduled to be released on Mar 25, 2022”. Please make sure it is available to be downloaded.

=> We thank the Reviewer for this comment. The single cell data of “Accession "GSE128879" is currently private until the study is accepted for publication. The data under this accession from GEO is for the scRNA-seq of in vitro morphine treated PBMCs. For the reviewer access, we created a secure token to access the study: gpqnaagorxqhvm

The raw scRNA-seq of LPS-treated and IFN β treated PBMCs from opioid-dependent and non-dependent control samples are available from dbGaP under accession phs000277.v2.p1: https://www.ncbi.nlm.nih.gov/projects/gap/cgi-bin/study.cgi?study_id=phs000277.v2.p1

12. Can you create a website or web portal so that readers can browser the data and results? It will also be a good resource for the folks in this community.

=> We thank the Reviewer for this recommendation. We have created studies for all three scRNA-seq datasets (accessions SCP587, SCP589, SCP591) through Single Cell Portal, which is currently private until the paper is accepted for publication. We have added a description of the Single Cell Portal studies in the Data Availability statement (page 11). For the Reviewer access, we have provided private access to all study results using the login credentials, username: singlecellpbmccopiate@gmail.com and password: Morphine123!

Reviewers' comments:

Reviewer #1 (Remarks to the Author):

All concerns have been adequately addressed.

Reviewer #2 (Remarks to the Author):

The authors have addressed my concerns and I recommend this paper for publication.

Simmie Foster

Reviewer #3 (Remarks to the Author):

Thanks for addressing my comments, especially on trying different ways for mixed linear modeling, and performing the microscope imaging validations. All the updates look good to me.

It is great to share the data through GEO, dbGap, and Broad Single Cell Portal. I tried to login the Single cell Portal using the given login username and password, but a phone message code is required. So I couldn't see the website to further examine the data for now. Can you fix this? That would be great to review some data on the website to see if the results are consistent with what has been written in the manuscript.

One minor suggestion: can you specify if you performed one-tailed or two-tailed T-test for your testings, e.g. Fig 2 and Fig 3.

Thanks
Fan

Reviewers' comments:

Reviewer #1 (Remarks to the Author):

All concerns have been adequately addressed.

We thank Reviewer 1 for their support of our work.

Reviewer #2 (Remarks to the Author):

The authors have addressed my concerns and I recommend this paper for publication.

We thank Reviewer 2 for their enthusiasm for our work and recommendation for publication.

Reviewer #3 (Remarks to the Author):

Thanks for addressing my comments, especially on trying different ways for mixed linear modeling, and performing the microscope imaging validations. All the updates look good to me.

It is great to share the data through GEO, dbGap, and Broad Single Cell Portal. I tried to login the Single cell Portal using the given login username and password, but a phone message code is required. So I couldn't see the website to further examine the data for now. Can you fix this? That would be great to review some data on the website to see if the results are consistent with what has been written in the manuscript.

One minor suggestion: can you specify if you performed one-tailed or two-tailed T-test for your testings, e.g. Fig 2 and Fig 3.

We apologize for the login issue for Single Cell Portal. We have resolved the issue and have updated the login credentials to:

Username: scpbmcreviewer@gmail.com

Password: Morphine123!

All three scRNA-seq datasets (accessions SCP587, SCP589, SCP591) will be available to look at under the My Studies tab in Single Cell Portal.

In addition, we performed a two-tailed T-test for our statistical testing in Figure 2 and Figure 3. We included this detail in the figure legends of Figure 2, 3 and Supplementary Figures 8-12, 15, 17-21 to make this clear.

REVIEWERS' COMMENTS:

Reviewer #3 (Remarks to the Author):

Thanks for fixing the issue of the single cell portal. It looks great!
No further comments.